# Improving Human Health with Milk Fat Globule Membrane, Lactic Acid Bacteria, and Bifidobacteria

**DOI:** 10.3390/microorganisms9020341

**Published:** 2021-02-09

**Authors:** Erica Kosmerl, Diana Rocha-Mendoza, Joana Ortega-Anaya, Rafael Jiménez-Flores, Israel García-Cano

**Affiliations:** Department of Food Science and Technology, The Ohio State University, Columbus, OH 43210, USA; kosmerl.3@buckeyemail.osu.edu (E.K.); rochamendoza.1@osu.edu (D.R.-M.); ortegaanaya.1@osu.edu (J.O.-A.)

**Keywords:** gut microbiota, infant formula, milk fat globule membrane, dairy foods, interactions

## Abstract

The milk fat globule membrane (MFGM), the component that surrounds fat globules in milk, and its constituents have gained significant attention for their gut function, immune-boosting properties, and cognitive-development roles. The MFGM can directly interact with probiotic bacteria, such as bifidobacteria and lactic acid bacteria (LAB), through interactions with bacterial surface proteins. With these interactions in mind, increasing evidence supports a synergistic effect between MFGM and probiotics to benefit human health at all ages. This important synergy affects the survival and adhesion of probiotic bacteria through gastrointestinal transit, mucosal immunity, and neurocognitive behavior in developing infants. In this review, we highlight the current understanding of the co-supplementation of MFGM and probiotics with a specific emphasis on their interactions and colocalization in dairy foods, supporting in vivo and clinical evidence, and current and future potential applications.

## 1. Introduction

The milk fat globule membrane (MFGM) is the structure comprised of lipids, membrane-associated proteins, and carbohydrates in the form of glycoconjugates, which surround every fat globule secreted in milk during lactation of mammals. The beneficial impacts of the bovine MFGM have been well-established [1]. In recent years, infant formula companies have begun to include MFGM ingredients in their formulations for its effects on cognitive development and gut maturation [2,3]. Furthermore, the focus on understanding how MFGM affects the gut microbiome of developing infants is gaining momentum. The current evidence demonstrates that MFGM supplementation increases the abundance of beneficial bacteria, such as bifidobacteria and some species of lactic acid bacteria (LAB), which is associated with a reduced risk of several diseases, including type I and type II diabetes, hepatitis B, and obesity (previously reviewed in [1]). However, the mechanism by which the MFGM imparts its benefits is much more poorly understood. In addition to the benefits of MFGM consumption, probiotics and their positive outcomes go beyond intestinal and immunological health by also directly impacting cognitive function. In this review, we aim to concisely summarize the current research on the MFGM and its interactions with probiotics both in foods and the gastrointestinal tract and discuss the potential mechanisms involved with their interactions, as well as the synergistic effects of co-supplementation.

### 1.1. The Milk Fat Globule Membrane

#### 1.1.1. Structure and Origin

To comprehend the complex structure and composition of the MFGM, it is necessary to understand its origin. The MFGM originates in the secretory cells of the mammary gland alveoli (Figure 1). Once the fat globules are synthesized in the rough endoplasmic reticulum (ER), they are coated with a first layer of phospholipids before being released to the cytoplasm [4]. They migrate to the apical part of the cell, where they are enveloped with an additional bilayer of phospholipids derived from the cell membrane and then expelled into the alveolar lumen [5]. During secretion, an additional layer that is electron density rich forms between the outer bilayer and the monolayer, which houses cytosolic proteins xanthine dehydrogenase/oxidase (XDH/XO) and the FABP family [6,7], as well as ADPH which is embedded in the phospholipid monolayer (Figure 1, insert). XDH/XO is colocalized with a small extracellular region of integral membrane protein BTN from the moment they are synthesized in the apical part of the secretory cells, as observed experimentally in mouse models [8]. The role of the MFGM during lactation is one of active participation in the secretion of fat globules. After lactation, it prevents the coalescence of fat droplets by maintaining the milk emulsion and serving as a barrier to avoid lipase and other enzymatic activity that could disrupt the globules [9].

#### 1.1.2. Composition

The gross composition of the MFGM varies greatly as a result of factors such as the physiology of the mammal, isolation and purification methods, processing of the milk, and even the analytical techniques used to quantify the molecular components (Table 1). In general, the concentration of MFGM in cream ranges between 3.0–3.9 g/L and is composed primarily of specific lipids and membrane-associated proteins that, together, account for over 90% of the MFGM dry weight.

As shown in Table 1, the MFGM is rich in lipids, the predominant class being milk phospholipids (MPLs) (Table 2). Phospholipids (PLs) are ubiquitous molecules in membranes and vesicles, and the MFGM possesses a unique mixture not found in plant-based lecithin. Phosphatidylethanolamine (PE) is usually found as an important constituent of nervous and brain tissue and, in the MFGM, constitutes up to 46% of the total MPLs [14], whereas phosphatidylcholine (PC) represents up to 38%, and its function is to maintain the permeability of the membrane due to its flexible morphology [15,16]. Phosphatidylinositol (PI) and phosphatidylserine (PS) are present in lower concentrations in milk and are anionic PLs distributed asymmetrically throughout the MFGM [17]. Sphingomyelin (SM) is the main sphingolipid in the MFGM due to its high concentration (about 30–40%), and it is known to have many beneficial effects on human health, such as improved neuronal development in infants and the protection of neonates from bacterial infections [18,19,20]. Structurally, SM forms lipid-ordered domains that also contain cholesterol (Figure 1B). This cholesterol appears to be involved not only in the topography and fluidity of the MFGM but, also, in biological functions such as localization and insertion of MFGM proteins with transmembrane domains [21,22,23]. The MFGM lipids are arranged asymmetrically in such a specific way that SM and PC are located on the outside of the bilayer while PE, PS, and PI are found inside the membrane [15]. MFGM glycolipids are almost exclusively located in the outer layer [24].

The composition of the MFGM components varies greatly between mammalian species. Even though the structure and biosynthesis are similar, the profile of MPLs is different, as seen in Figure 2. The composition of human MPL in comparison to bovine milk differs mainly in the concentration of SM and PS—human milk provides a higher percentage of SM, whereas bovine milk has a higher percentage of PS. The composition of MPLs from donkey milk is extremely different from that of milk of all other species, particularly due to the high values of PE and the low values of SM.

Gangliosides are glycolipids involved in neuronal development and immunological adaptation in neonates and, hence, are very important in human nutrition. GD3 is the main ganglioside in bovine and human MFGM, followed by GM3, although the concentrations are different [25].

The MFGM possesses a rich variety of proteins as well, which are all membrane-associated and account for about 25% of the mass [34]. The protein profile is vast, and it varies between species. Human, bovine, and caprine MFGM possess 1104, 632, and 137 different proteins, respectively [35]. However, a group of eight proteins are usually the most abundant in all species. Figure 1B depicts this group of predominant MFGM proteins, as well as their locations in the different layers, and their orientation and association with phospholipids, carbohydrates, and other proteins. Table 3 summarizes their molecular size and, more importantly, their observed effects on human health.

There is a lot to say about the MFGM proteins given their complex compositions. In addition to the predominant group shown in Table 3, hundreds of other proteins in minor concentrations are also found, and according to proteomic studies, we know that the distribution varies greatly upon the stage of lactation, environmental conditions, the method of extraction, and the processing of the source containing MFGM. These minor proteins have important functions in biological processes, such as antioxidant and detoxification properties, and molecular functions such as binding [35]. Finally, it is important to note that, according to the techniques used to isolate MFGM and process milk, whey proteins (α-lactalbumin, β-lactoglobulin, immunoglobulins, and lactoferrin) are often found associated with the MFGM fraction [34].

### 1.2. Lactic Acid Bacteria (LAB) and Bifidobacteria

With this review, we aim to showcase the interactions between the MFGM and probiotic bacteria, but first, we must highlight some of the notable features of these bacteria to better understand their behaviors in complex systems. The two major groups of probiotic bacteria are LAB and bifidobacteria. LAB are an important group of microorganisms of great relevance for the food, clinical, and agricultural industries. Their main application is in the preservation of foods by the production of lactic acid through fermentation. According to the current taxonomic classification, the LAB group belongs to the phylum Firmicutes, class Bacilli, and order Lactobacillales, which comprises six families: Aerococcaceae, Carnobacteriaceae, Enterococcaceae, Lactobacillaceae, Leuconostocaceae, and Streptococcaceae [43]. In general, LAB are Gram-positive, non-spore-forming, aerotolerant, acid tolerant, rod- or coccus-shaped bacteria that produce lactic acid and other organic compounds as a final product of carbohydrate catabolism, contributing to the taste, texture, and final aroma of foods that contain them [44]. These bacteria are usually nonmotile and present a complex, nutritionally fastidious requirement of vitamins, nutrients, and amino acids for growth. LAB are distributed in nature in nutrient-rich habitats, including niches such as food (dairy, fermented meat, cereals, fruits, and vegetables) and in human and animal cavities (mouth, genitals, respiratory, and intestinal tracts).

Although LAB are already a diverse group of bacteria, they are further diversified by containing circular and linear plasmids that give them the ability to survive in different environments and contain additional genes that alter their metabolism. These include genes for the metabolism of amino acids, citrate, carbohydrates, and proteins, as well as genes involved in the production of bacteriocins, exopolysaccharides (EPSs), and pigments; defense mechanisms against phages; and resistance to antibiotics and heavy metals [45]. LAB are classified based on their physiological and biochemical characteristics, such as their growth at different temperatures (15–45 °C), ability to grow in high salt concentrations (6.5 and 18% NaCl), tolerance to alkali or ethanol, cell wall or fatty acid compositions, and capacity for acid production during carbohydrate fermentation. Based on their metabolic products, LAB can be classified as homofermentative (i.e., *Lactococcus* and *Streptococcus*) or heterofermentative (i.e., *Leuconostoc*, *Wiessella,* and some *Lactobacillus*), which produce two lactate molecules per glucose via the Embden-Meyerhof-Parnas pathway or lactate, ethanol, and carbon dioxide from glucose via the 6-P-gluconate/phosphoketolase pathway, respectively [46]. 

Another group of bacteria, called bifidobacteria, are commonly associated with LAB in fermented foods but are distinct in their physiological, biochemical, and metabolic characteristics. The genus *Bifidobacterium* belongs to the Bifidobacteriaceae family, order of Bifidobacteriales, and belong to a branch of the phylum Actinobacteria [47]. Currently, this genus contains 80 recognized taxa, 95 species, and 18 subspecies (https://lpsn.dsmz.de/genus/bifidobacterium) divided into six phylogenetic groups: *B. adolescentis, B. longum, B. pullorum, B. asteroides, B. pseudolongum,* and *B. boum* [48]. Bifidobacteria are Gram-positive, non-spore-forming, nonmotile, mostly anaerobic, and generally rod- or bifid-shaped bacteria. Their optimal growth temperature ranges from 37 to 41 °C. These bacteria can produce acetic and lactic acids from the fermentation of glucose, galactose, and fructose without generating carbon dioxide. Unique to bifidobacteria, carbohydrate fermentation occurs through the fructose-6-phosphate phosphoketolase pathway, also called the Bifid shunt. It is through this pathway that bifidobacteria are well-recognized for their utilization of human milk oligosaccharides (HMOs) for growth and gut microbiome development in infants [49]. Their main niche is the intestinal tract and mouth of humans or animals, although they have also been isolated from birds and insects, sewage, fermented milk, and blood. 

Bifidobacteria and some LAB are generally considered nonpathogenic and are commonly used as probiotics in fermented products and food supplements. Probiotics are defined as “living microorganisms that when administered in adequate proportions can confer benefits on the health of the host” [50]. However, the applications of bifidobacteria and LAB differ because of their differences in metabolism and physiology. For example, bifidobacteria have certain growth disadvantages in fermented products, including dairy products, compared to LAB. They grow and acidify poorly in cow’s milk; have low proteolytic activity; and require longer fermentation periods, anaerobic conditions, and low redox potential to grow. Some of these challenges can be overcome through addition of substances such as ascorbic acid or cysteine to reduce the redox potential and promote growth [51]. In addition, some species can survive the acidic environment of the stomach and duodenum and the presence of bile salts and pancreatic juices, but these are very aggressive conditions for bifidobacteria, and their proportions diminish over time. LAB are commonly used as starter cultures for food fermentation due to the metabolites that they produce, such as lactic acid and proteins with antibacterial activity. They prevent the decomposition of food and the growth of pathogenic microorganisms.

Both LAB and bifidobacteria colonize the gastrointestinal tract of the host by adhering to intestinal cells, exhibiting a resistance to host barriers [52]. The adhesion of bacteria in the intestine is related to the presence of mucins whose functions are to lubricate and protect the epithelial cells, increasing the adherence of LAB and bifidobacteria while concurrently excluding pathogenic bacteria. Pathogen exclusion occurs through various mechanisms. For example, *Lactobacillus* can produce substances with antimicrobial activity with an inhibitory effect against pathogenic enteric bacteria [53]. LAB and bifidobacteria are essential for the health of the host by their direct involvement in metabolism, digestion, and preservation of the immune system [54]. Both LAB and bifidobacteria are found as part of the human milk microbiome. Although the exact composition of bacteria varies between mothers, the predominant species in breast milk include *Ligilactobacillus salivarius*, *Limosilactobacillus fermentum*, *Lactobacillus gasseri*, *B. breve*, *B. adolescentis*, and *B. longum* subsp. *infantis* [55]. LAB, through delivery in infant formulas, have been shown to promote health through mitigation of several conditions in infants. For example, *L. fermentum* CECT5716 has been shown to reduce the risk and duration of diarrhea [56], *Limosilactobacillus reuteri* DSM17398 has been shown to aid in colic management [57], and *Lacticaseibacillus paracasei* CBA L74 is protective against colitis and pathogen infection [58]. In addition, bifidobacteria is one of the predominant groups associated with a healthy human microbiota. They rapidly colonize the intestine of infants and are transmitted directly from the mother to the infant to form part of their microbiota. Bifidobacteria found in infants modulate their metabolism toward degradation of oligosaccharides present in breast milk, while those found in adults mainly degrade complex carbohydrates from the diet, establishing an interaction between bifidobacteria and other microorganisms present in the gastrointestinal tract [47]. The predominant species in infants are *B. breve, B. infantis,* and *B. longum*, while *B. catenulatum, B. longum,* and *B. adolescents* are the species commonly found in the gastrointestinal tract (GIT) of adults [51]. Their presence has been attributed to the promotion of health benefits such as improving lactose digestibility, synthesis of vitamin B, facilitating calcium absorption, preventing diarrhea, reducing cholesterol levels, production of vitamins, immunostimulation, and anticarcinogenic effects [59].

## 2. Interactions of Bifidobacteria, LAB, and MFGM in Dairy Food Matrixes

Milk and dairy products are the most popular matrices for the dietary delivery of LAB and bifidobacteria to humans, but little is known about how the MFGM interacts with and impacts bacterial cells. Moreover, no mechanism has yet been fully described to explain this interaction. The knowledge of the latter could very well contribute to the development of more efficient dairy foods, with an increased impact on human health. Since the middle of the last century, the inclusion of LAB in fermented foods has generated a revolution in the processing, production, and consumption of foods. Since that time, several starter cultures have been developed for fermentation and direct inoculation in food matrices. A starter culture is defined as a group of microorganisms that are inoculated directly into the food matrix with the intention of providing desirable changes in the final product [60]. One application of the interactions between LAB and MFGM is their utilization for the cryoprotection of stock and starter cultures. Bacterial freezing processes decrease the cell viability, damage the bacterial membrane, and reduce their functionality. The supplementation of 0.5% MPLs in acid whey-based media has been reported to protect LAB from freeze/thaw cycles [61]. Starter cultures also induce changes in a final product, including improved sensory and nutritional properties and compositional changes for the preservation of foods, as well as an added economic value [62]. The reactions amounting to these changes are directly related to the presence and composition of the carbohydrate, protein, and lipid components [63]. 

The interaction between MFGM and LAB and bifidobacteria is an adhesion phenomenon driven primarily by the bacterial surface properties. It is a very complex relationship that depends on many factors, including, but not limited to, strain genotype and phenotype and bacterial biochemistry, such as environmental conditions, cell viability, and metabolic activity. It is also impacted by the dairy matrix composition, water content, and processing strategies used upon the foods. Hence, there is no current set of rules or guidelines to predict whether the interaction will occur in all food matrices or under what mechanism; it should be reviewed case-by-case. In general, bacterial adhesion occurs in two steps: the first contact with the substrate is nonspecific and governed by reversible interactions such as electrostatic, van deer Waals, and Lewis acid–base. This step is followed by nonreversible and specific interactions involving adhesion factors, complementary receptors, surface appendages, teichoic and lipoteichoic acids, and EPSs from the cell envelope [64,65]. Given these phenomena, we should expect an attachment of LAB and bifidobacteria to the MFGM to occur in the same manner.

In dairy products, we now know that LAB and bifidobacteria are preferably associated with the fat/protein interfaces in cheeses at first, and after a period of aging, they have been found embedded in the MFGM or inside the fat globules when they exert their lipolytic activity [66,67,68,69]. Recently, a 45.9-kDa phosphoesterase with activity toward PL was isolated and characterized from *Pediococcus acidilactici* [70], which indicates a variety of lipolytic enzymes found in LAB and their affinity not only to milk fat but to MPL as well. Other research suggests that LAB and bifidobacteria may have additional activity beyond lipolysis of the MFGM components. For example, lactobacilli isolated from cheese has been shown to grow and utilize the monosaccharides present in MFGM glycoconjugates [71], which may alter the flavor development in cheeses [72].

In fresh cream, the direct association of LAB to fat droplets was informally reported in 2008 [73], and ever since, strains such as *Lactococcus lactis* ssp. *lactis* subv. Diacetylactis [74] and *L. reuteri* SD2112 and T1 have been shown to attach directly to the MFGM in the surface of fat globules (Figure 3). Moreover, Brisson and colleagues were able to establish that the mechanism of adhesion occurs through specific connections with the bacterial surface proteins [75].

Simple models have shown that many strains of LAB directly bind to MFGM components. Guerin and colleagues used atomic force microscopy to identify that *Lacticaseibacillus rhamnosus* GG (ATCC 53103) binds MFGM isolated from raw cream using functionalized probes, and moreover, they identified that the adhesion occurs through the interaction of SpaCBA pili in the bacterial cell wall [76].

It is clear that many strains of LAB, in fact, interact with the MFGM; however, the identification of the specific components from the MFGM that interact with these bacteria are not well-known to date. We can hypothesize, based on the literature, that LAB are prone to bind primarily to glycans of the MFGM proteins, such as PAS6/7 and mucins, due to their high degree of glycosylation (approximately 10% and 60%, respectively) [40,77,78,79]. MUC1 is resistant to gastrointestinal digestion, likely due to its sugar moieties, including fucose, galactose, mannose, N-acetylgalactosamine, and N-acetylglucosamine, among other sugars, which sterically protect it from proteolytic degradation [79,80]. PAS6/7 has been previously reported to retain its ability to bind bacteria due not only to its glycosylation but, also, its association with lipids and, specifically, PS [40,81]. A study used the sucrose density gradient to show that six strains of *L. reuteri* isolated from various dairy products bind MPLs from the MFGM [82]. *L. reuteri* showed greater interactions with MPLs compared to *Pediococcus lolii*, suggesting that these interactions can vary at the genus and species levels. In addition, it was found that some of the strains contained an increased relative expression of surface-binding promoting genes *CmbA*, *Cnb*, and *MapA* with the addition of 1% (*w*/*v*) of MPL. Mucus adhesion-promoting protein (MapA), the collagen-binding protein (Cnb), and a putative sortase-dependent cell and mucus binding protein A (CmbA) have been shown to promote the adhesion of bacteria to intestinal epithelial cells [83,84,85,86]. 

In our research group, through transmission electron microscopy (TEM), we identified that MPLs from the MFGM in simple models bind directly to the surface of some *Lactobacillus* (Figure 4(A1,B1)) in a strain-dependent manner during cell growth rather than internalizing them for further metabolism. This, in turn, affects the parameters of the bacteria, such as the growth rate; cryotolerance; surface hydrophobicity; ζ-potential; and adhesion properties such as expression of adhesion factors and mucus-binding proteins [82,86,87] (Ortega-Anaya, J., Ohio State University, Columbus, OH, USA; personal communication, 2020). In complex matrices such as milk and dairy products, the interaction, as well as the outcome, is expected to be more complex. 

## 3. Evidence of Improved Health with Combined Probiotic and MFGM Supplementation

Although the exact mechanism of interaction between probiotics and the MFGM has not been fully elucidated, the evidence of the synergistic effects upon the combined supplementation of these functional ingredients suggests beneficial outcomes in both in vitro and in vivo studies, including studies in infants (summarized in Table 4).

### 3.1. Bacterial Survival and Adhesion

Survival through gastrointestinal transit is considered an optimal trait for probiotic efficacy and bioactivity; however, probiotics are challenged with, and must overcome, the harsh physical and chemical environments of the GIT [95]. Probiotic survival is dependent on a tolerance to acidic pH and bile salts, bile salt hydrolase activity in the GIT, and the duration of exposure to these stressors [96]. Among the strategies used to boost probiotic survival in the GIT is the combination of probiotics with other nutritional components, such as prebiotics [97], or the use of milk as a delivery vehicle [98]. One study examined the potential for MFGM-10, an ingredient derived from whey protein concentrate (Lacprodan MFGM-10, Arla Food Ingredients, Viby, Denmark), to enhance the survival of *L. rhamnosus* GG (LGG) [87]. As little as 2.5-g/L MFGM-10 significantly improved the survival of LGG to 30-min exposure of 0.5% bile conditions. In contrast, non-MFGM enriched whey protein concentrate (WPC) was unable to protect LGG from bile stress, demonstrating that increased survival is due to a specific property of the MFGM. Autoclaving had no effect on the protective activity of MFGM, indicating that this property is not thermolabile under these conditions. They also measured the survivability of LGG by cecum and fecal counts in mice, confirming the protective activity of MFGM in vivo. Mechanistically, bile stress upregulated EPS production of the bacteria. In the presence of MFGM, EPS production was downregulated more similarly to control the bacteria that lacked exposure to bile. EPSs have been shown previously to increase biofilm formation, increase intestinal adhesion, and influence the tolerance of the host immune system, all of which are important characteristics of probiotic activity [87,99]. Another study examined the contribution of the *Lactobacillus johnsonii* FI9785 EPS to facilitate interactions with the host via adhesion, biofilm formation, and stress resistance. These authors found a correlation of decreased EPS production with increased adhesion to chicken gut explants and a greater susceptibility to stress by bile salts and antibiotics. Therefore, it is possible that there is a trade-off between adhesion and survival in the GIT. EPSs also play a role in immune tolerance toward commensal bacteria and protection from pathogens [100]. This immune tolerance may be a plausible reason for why the delivery of probiotics in dairy matrices has been so successful—aside from the fact that LAB are nutritionally fastidious. Taken together, these findings demonstrate the direct effect of MFGM on bacterial metabolism and gene expression, which further raises the questions: what constituents of the MFGM aid in the regulation of EPS production, and how do these changes in survival impact adhesion in the gut? 

Adhesion is a prerequisite for colonization that has sparked both biophysical and in vitro biological investigations into the influence of MFGM or MPLs on probiotic adherence to intestinal mucus or cells, as measuring colonization in vivo is challenging. Using a quartz crystal microbalance with dissipation (QCM-D) to measure the binding kinetics of LAB grown in the presence of MFGM-derived MPLs, our group found that MPLs increased the affinity for certain LAB species, including *Lacticaseibacillus casei* OSU-PECh-C and *Lactobacillus acidophilus* Musallam 2 to a gold sensor that acts as a universal binding surface [101]. Other strains, including *L. plantarum* subsp. *plantarum* TW14-1 and *L. delbrueckii* OSU-PECh-3, exhibited weaker interactions with the sensor. Interestingly, when these bacteria were applied to the Caco-2/HT29-MTX coculture, *L. delbruekii* and *L. casei* displayed increased adhesion in the presence of MPLs, suggesting MPLs increase the affinity for intestinal cells in a strain-dependent manner in vitro [88]. Using more traditional culture methods, Rocha-Mendoza et al. [86] found that 0.5% MPLs significantly increased the adhesion of *P. acidilactici* OSU-PECh-L, *L. plantarum* OSU-PECh-BB, and *L. reuteri* OSU-PECh-48 to in vitro Caco-2 cells, but MPLs had no significant change on the adhesion of the four other strains tested. In contrast, the MFGM extract derived from butter serum significantly reduced the number of adherent bacterial cells of the commercial probiotic LGG to Caco-2 TC7 cells [76]. Furthermore, high-throughput assays used to assess the effects of several dairy fractions on the adhesion of *B. longum* subsp. *infantis* ATCC 15697 to HT-29 cells showed that Lacprodan PL-20 (an MPL-rich ingredient from Arla Food Ingredients) and the buttermilk fraction significantly reduced the bacterial adhesion. MFGM did not alter the bacterial adhesion. These authors did not, however, report whether they used a mucus-producing clone of HT-29 cells, such as HT29-MTX-E12. As the mucus layer in the GIT serves as the key site of interaction between the host and microorganisms, it is necessary to conduct further research that includes this important structure to better understand the relationships between MFGM, probiotic survival, and adhesion. 

With these ideas in mind, the next generation of studies must address whether the combination of MFGM and probiotics are capable of influencing colonization in vivo. However, these studies must additionally overcome the challenges associated with these measurements, as the fecal microbiome is not well-representative of the colonized microbiome in the GIT [102], and differences in microbiomes between species are not easily translatable [103]. Furthermore, Zmora et al. [102] reported that the colonization of probiotics in the GIT of adult humans is limited by individual host colonization resistances resulting from the resident microbiome, as the degree of probiotic enrichment varied between significantly between individuals.

### 3.2. Nutrient Absorption, Mucosal Immunity, and Gut Barrier Function

The mucosal immune system orchestrates the body’s ability to absorb nutrients and tolerate commensal bacteria while simultaneously providing protection from pathogens by initiating (and later resolving) an immune response. Components such as the gut-associated lymphoid tissues (GALT), Peyer’s patches, mucosal immune cells, and commensal bacteria work together to produce antimicrobial peptides, cytokines, and chemokines that maintain the gut homeostasis [104]. It has been reported that a milk fermentation containing SM and two LAB (*L. delbrueckii* subsp. *bulgaricus* 2038 and *Streptococcus thermophilus* 1131) increases the absorption of this phospholipid in murine models [90]. As previously discussed, SM is regarded for its function in regulating inflammation [105], promoting neurodevelopment and cognitive function [106,107], and improving the skin barrier function [108], among others.

It is well-documented that dietary composition and, specifically, milk components can directly influence the ability of the mucosal immune system to act in an appropriate and efficient manner and influence the proper development in infants [109,110]. A formula containing milk fat stabilized by MFGM fragments improved the markers for mucosal immune development, including IFNγ secretion and intestinal microbiota composition in neonatal piglets [109]. Specifically, the formula led to an increase in *Proteobacteria* and *Bacteroides* and decrease in *Firmicutes* compared to a formula containing vegetable fat stabilized by a mixture of proteins. Although the microbial composition changes in piglets do not translate well to infants, the authors did observe a similar increase in *Parabacteroides*, which is a characteristic of breastfed infants [103]. Others have also shown that MFGM in infant formula can modify the fecal microbiota and metabolism of infants through changes in serum metabolites [111,112]. These data and others (reviewed in [34]) suggest that, individually, MFGM plays an important role in mucosal immunity. However, a co-supplementation with probiotics is purported to have greater synergistic effects on the mucosal immunity [92,113].

Specifically, one patent (WO2004/112509A2) describes an infant formula containing MFGM and, at minimum, one probiotic bacterial strain that acts synergistically to drive gut barrier maturation more closely to that of breast milk-fed infants [113]. These probiotic strains include *Bifidobacterium adolescentis* CNCM I-2168 (Bad4), *B*. *longum* CNCM I-2169 (Bl28), *B*. *longum* CNCM I-2170 (Bl29), *L. paracasei* ST11, *Streptococcus thermophilus* TH4, *B*. *animalis* subsp. *lactis* BB-12 (BB-12; ATCC27536), and *B*. *longum* BB536. The authors proposed the use of a probiotic cocktail in combination with MFGM or other bioactive components (i.e., human milk oligosaccharides, gangliosides, and sweet or acid whey), as each microorganism has differences in survival throughout the GIT. Another patent (WO2011/069987) embodies an infant formula that contains MFGM and probiotics that can be used as a therapeutic for disease prevention and to help drive normal gut development in infants [92]. The invention shows interactions between the MFGM and the commercial probiotic *B. animalis* subsp. *lactis* BB-12 (BB-12), which may help protect the probiotic during transit to the gut, where they can then modulate the immune system. Using an NF-κB reporter epithelial cell line, the combination of MFGM and BB-12 showed a synergistic effect on NF-κB activation in response to the challenge with an endotoxin. The combination decreased NF-κB activation to a greater extent than MFGM or BB-12 alone. In addition, four weeks of combined MFGM and BB-12 consumption led to significantly higher IgA-secreting mucosal B cells in mice compared to BB-12 or MFGM alone. Secretory IgA produced by these cells has been well-studied for its role in mediating a tolerance to commensal gut microbes [114]. Together, these data show enhanced immune maturation and education and reduced inflammation when supplementations with MFGM and BB-12 are combined. Combined supplementation may offer benefits for developing infants, as well as for the prevention and treatment of gut-related diseases.

### 3.3. Neurodevelopment and Cognitive Function

During postnatal life, the brain rapidly undergoes changes in size, reaching 64% of its adult size within the first three months of life [115], as well as vast changes in neural networks stemming from increased exposure to enriched environments and experiences [116]. It is well-documented that nutrition and diet can also influence neurodevelopment in the postnatal period [106,117,118]. Moreover, MFGM and MFGM-derived MPLs alone have strong supporting evidence of the positive effects on cognitive function and brain development [1,119,120]. For example, an infant formula containing MFGM improved the reflex development and altered the brain lipid composition in rats with a pup-in-a-cup model more similarly to mother’s milk [121]. MFGM in formula has also been reported to improve T-Maze scores in growth-restricted rats [122], improve stress-induced disruptions in rapid eye movement (REM) sleep [123], and improve infant scores using the Bayles Scale of Infant Development [124]. Combining MFGM with other bioactive ingredients, such as probiotics, may provide additional benefits throughout development, although the mechanisms have not yet been fully elucidated.

As part of a larger study called COGNIS, the effects of a functional infant formula containing MFGM components, symbiotics (a combination of prebiotics and probiotics), and other bioactive components on visual function as a measure of neurodevelopment was evaluated in infants [93]. The authors found that infants fed the functional formula had similar response rates to the breastfed group by the measure of cortical visual-evoked potentials at 12 months of age. The standard formula-fed infants had a significantly lower proportion of responses, suggesting that supplementation with these bioactive ingredients within the first year of life may promote neurodevelopment more similarly to breastfed infants. Ultimately, these results are promising for the combined supplementation of these bioactive ingredients; however, it is still necessary to better define which bioactive attributes are leading to these effects, as the functional formula contained other ingredients, such as prebiotics (fructooligosaccharides and inulin), as well long-chain polyunsaturated fatty acids (LC-PUFA). Another part of the COGNIS study investigated the effects of the same functional infant formula on long-term language development at four years of life [94]. The Oral Language Task of the Navarra-Revised (PLON-R) test was used to measure language development by categorizing kids into one of three categories: delayed, need to improve, or normal. The standard formula group was significantly more likely to fall into the delayed or need to improve categories than either the breastfed or functional formula groups. Additionally, the functional formula group showed significantly higher language use and oral spontaneous expression compared to the standard formula group. These data suggest that infant diets affect the long-term cognitive activity with respect to language development and that the diet containing both MFGM and probiotics narrowed the gap between formula-fed and breastfed infants.

## 4. Current Applications and Future Perspectives

The majority of research investigating the relationships between MFGM and probiotics occur in the area of infant formulas. MFGM research has long focused on the health benefits it provides, so logically, the idea of co-supplementations can be easily transitioned to this area. In infants, brain development and gut microbiota maturation are synchronized [125]. Although it appears that the co-consumption of MFGM and probiotics may hold additional developmental advantages, it is of great importance to better understand how this connection intertwines with the gut–brain axis, which has been shown to influence emotional, mental, and developmental health [126]. For example, the MFGM may influence not only the microbiota composition but, also, the metabolite production of the probiotics and resident gut microbiota—in turn, affecting the gut–brain axis [127]. Along with these ideas, we believe research in this area is important, as we have been left with several unanswered questions. At this point, it is still unclear whether the MFGM interaction with the bacterial surface facilitates the delivery of these probiotics to their target site of action or whether the MFGM creates an environment in the GIT that favors the probiotic species, alters their metabolism, and, ultimately, influences the gut microbiome. Relatedly, studies that further investigate the effects of MFGM or MPLs on bacterial surface biomolecules, such as EPSs and/or pili, in a variety of strains should be considered to uncover the mechanisms and, perhaps, the improvement of probiotic permanence in the GIT. As MFGM research continues to expand to other areas, including metabolic diseases [128], geriatrics and aging populations [129], and skin health [108], it would be no surprise to branch into these other areas as well. 

In this review, we addressed the unique and complex composition of the MFGM, including the presence of phospholipids and sphingolipids. LAB isolated from dairy products have been shown to produce lipases, esterases, and phospholipases [130]. The lipolytic system from LAB is considered weak in comparison to their proteolytic processes. Yet, there have been several reports of the production of lipases by LAB [70,131,132]. Moreover, in matured products, such as aged cheeses, LAB release intracellular lipases after cell disruption. In addition to the lack of detailed, mechanistic evidence of lipolytic activity at the MFGM, there is an urgent need for expanding our understanding of the probiotic activity on other components such as MFGM glycoconjugates and proteins. Through this improved understanding, we can expand the flavor profiles of fermented foods, find novel bioactive metabolites to benefit human health, and use these concepts to create novel functional foods.

## 5. Conclusions

The advancement of human health through a co-supplementation of MFGM and probiotics holds significant promise and calls for the further exploration of potential synergies and interplaying mechanisms. The current evidence supports a functional role of the MFGM to probiotic bacteria like that of prebiotics. However, the exact mechanism of interaction between probiotics and the MFGM has yet to be fully elucidated. It is becoming increasingly evident that the MFGM may alter the metabolism of probiotics to enhance their efficacy in the gut by promoting probiotic survival, boosting mucosal immune development, and acting upon the gut–brain axis for improved cognitive function. These studies warrant further investigation into the other related aspects of gut health, such as the influence of the combined supplementation of MFGM and probiotics on the gut microbiota and more. 

## Figures and Tables

**Figure 1 microorganisms-09-00341-f001:**
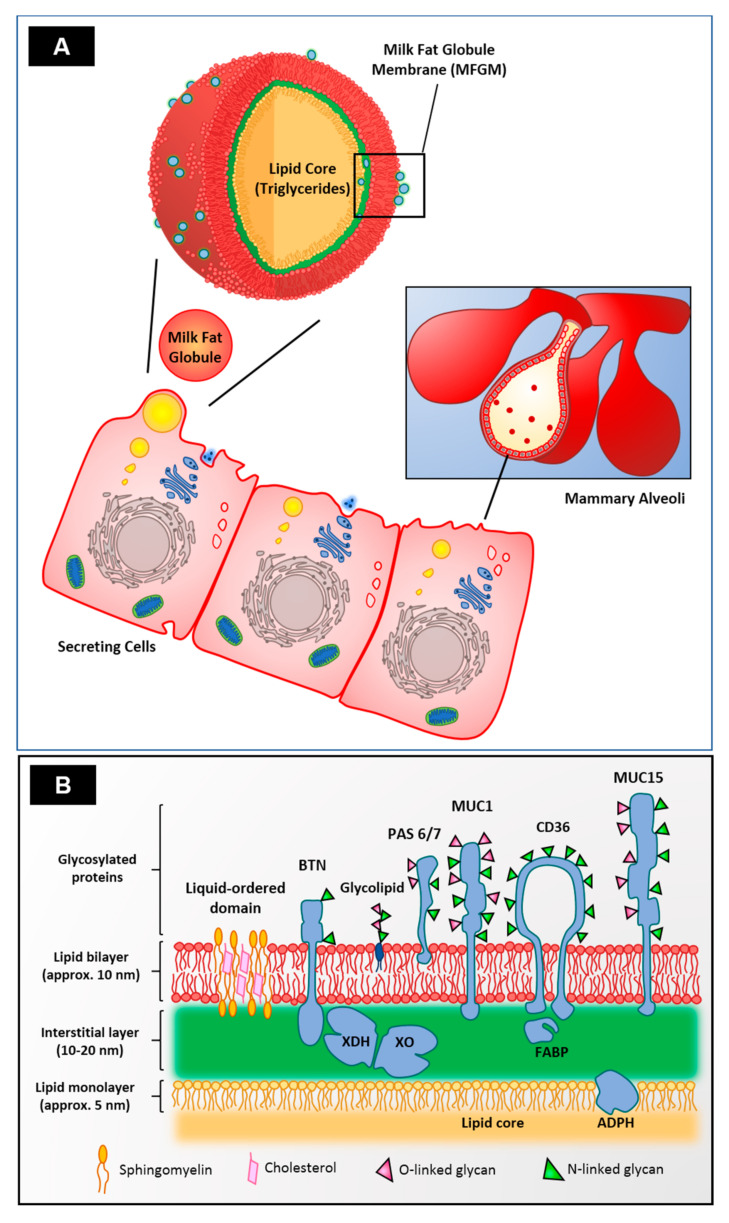
Schematic diagram of (**A**) the biosynthesis of fat globules during lactation and (**B**) the structure of the milk fat globule membrane (MFGM) where the different layers and their general compositions are depicted. BTN, Butyrophilin; PAS 6/7, Lactadherin; MUC-1, Mucin 1; CD36, Cluster of differentiation 36; MUC-15, Mucin 15; XDH/XO, Xanthine dehydrogenase/oxidase; FABP, fatty acid binding protein family; and ADPH, Adipophilin. Adapted with permission from Ortega-Anaya et al. (2020). Copyright 2020 Elsevier Ltd. [7].

**Figure 2 microorganisms-09-00341-f002:**
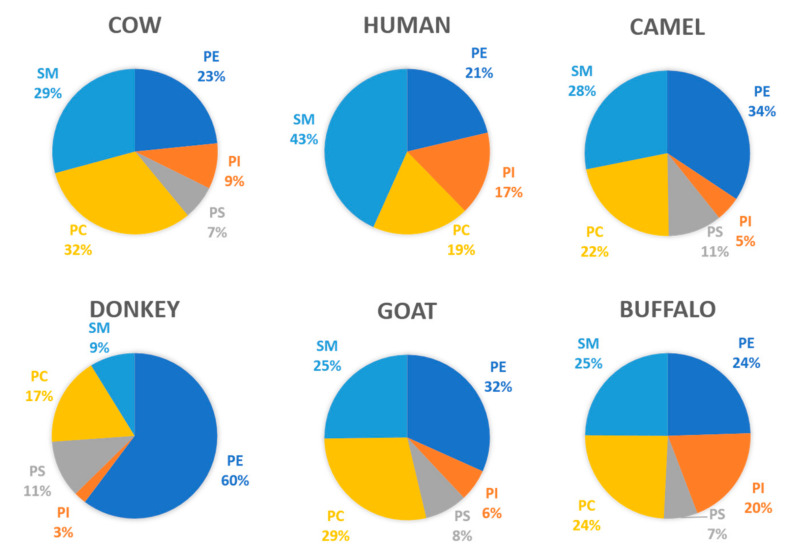
Variability of the milk phospholipid (MPL) profiles of different mammalian species. Data compiled from [28,29,30,31,32,33].

**Figure 3 microorganisms-09-00341-f003:**
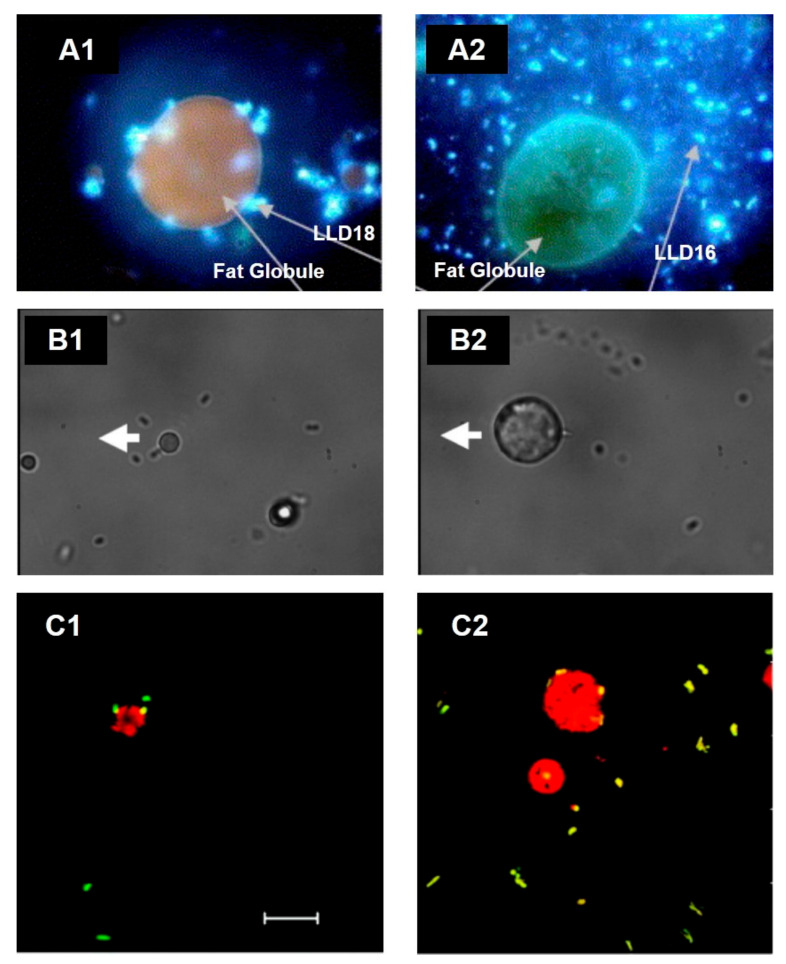
Association of lactic acid bacteria (LAB) with the MFGM in cream imaged by microscopy. (**A**) Fluorescence microscopy images of *L. lactis* ssp. *lactis* subv. Diacetylactis. (**A1**) Variant 18 (LLD18) is clearly attached to the MFGM, whereas (**A2**) variant 16 (LLD16), in comparison, is not. Adapted with permission from Ly et al. (2006) Copyright 2006 Elsevier Ltd. [74]. (**B**) Optical tweezer force images of *L. reuteri* SD2112 attached to the MFGM. (**B1**) Bacterial cells fixed to the cover slide, whereas (**B2**) shows the experiment when the MFGM is attached to the cover slide. The thick white arrow shows the direction of the microscope stage travel. Adapted with permission from Brisson et al. (2010). Copyright 2010 American Chemical Society [75]. (**C**) Confocal laser scanning images of (**C1**) *L. reuteri* SD2112 and (**C2**) *L. reuteri* T-1 in raw cream. Scale bar = 10 µm. Adapted with permission from Brisson et al. (2010). Copyright 2010 American Chemical Society [75].

**Figure 4 microorganisms-09-00341-f004:**
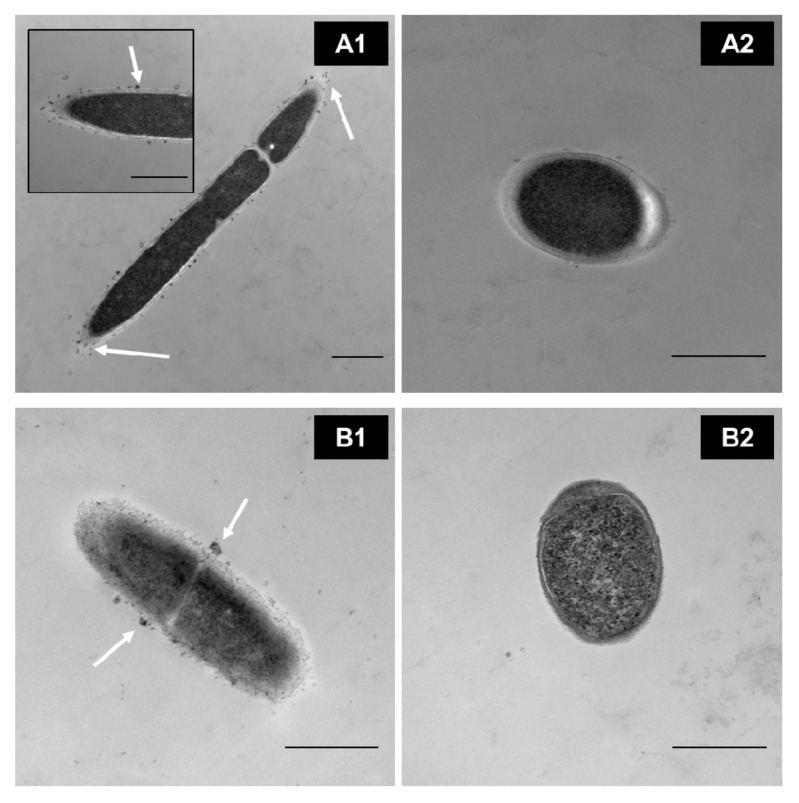
TEM images of washed cells grown in a defined medium supplemented with 0.5% of MPL. (**A1**) *Lactobacillus delbrueckii* vs. (**A2**) control. (**B1**) *Lactiplantibacillus plantarum* vs. (**B2**) its control. Scale bar = 500 nm. White arrows indicate MPL accumulation at the cell surface(Unpublished images, Ortega-Anaya (2020) Ohio State University, Columbus, OH, USA; personal communication).

**Table 1 microorganisms-09-00341-t001:** General composition of the milk fat globule membrane (MFGM). Data compiled from [10,11,12,13].

Component	Concentration Range (%)
Lipids	64–71.8
Protein	22.3–28
Glycoconjugates	10
RNA	Traces

**Table 2 microorganisms-09-00341-t002:** Composition of the MFGM lipids. Data compiled from [7,9,10,25,26,27].

Lipid Class	Fraction Content In %	Subclass
Polar Lipids		
Phospholipids	65	Glycerophospholipids - PE - PC - PS - PI Sphingolipids - Sphingomyelin (SM) ^a^ - Glucosylceramide (GlcCer) ^b^ - Galactoceramide (GalCer) ^b^ - Lactosylceramide (LacCer) ^b^
Gangliosides or glycolipids	5	- Monosialoganglioside 1 (GM1) - Monosialoganglioside 2 (GM2) - Monosialoganglioside 3 (GM3) - Disialoganglioside 1A (GD1A) - Disialoganglioside 1B (GD1B) - Disialoganglioside 2 (GD2) - Disialoganglioside 3 (GD3) - Trisialoganglioside (GT) - Trisialoganglioside 3 (GT3)
Neutral Lipids		
Di-acylglycerols	10	
Mono-acylglycerols	Traces	
Free fatty acids	10	
Sterols		
Cholesterol	10	
Liposoluble molecules		
Vitamins A, D, E and K	5	

^a^ Considered a sphingophospholipid. ^b^ Categorized as a neutral glycolipid and cerebroside.

**Table 3 microorganisms-09-00341-t003:** Composition of the predominant MFGM proteins. Data compiled from [36,37,38,39,40,41,42].

Protein	Content in g/100 g of Total Protein *	Molecular Mass (kDa)	Reported Effect on Human Health
Mucin 1 (MUC-1)	NA	250–450	Antiviral and antibacterial by preventing binding of pathogens to intestinal cells
Mucin 15 (MUC-15)	NA	94–100	Antiviral action
Xanthine dehydrogenase/oxidase (XDH/XO)	0.58	150–155	Bactericidal action by production of hydrogen peroxide and nitric oxide
Cluster of differentiation 36 (CD36)	0.18	77–78	Receptor for collagen and thrombospondin. Scavenger receptor for apoptotic cells
Butyrophilin (BTN)	3.35	66	Member of the immunoglobulin superfamily, adhesive protein, acts as a receptor and has a positive effect in the immune system. Co-inhibitor of T-cell activation
Adipophilin (ADPH)	0.007	52	Facilitates transport of triglycerides and fatty acids during fat globule synthesis
PAS 6/7 or Lactadherin	0.93	48–54	Adhesive properties with effect in the regulation of epithelial coagulation. Role in synaptic activity in the central nervous system (CNS) and protection against viral infection in the gut
FABP family	0.17	14–15	Transport of fatty acids

* In raw cream. NA: not available.

**Table 4 microorganisms-09-00341-t004:** Purported health benefits of combined probiotic and MFGM supplementation.

Ingredient	Microorganism(s)	Model	Experimental Design	Key Findings	Ref.
**Bacterial Survival and Adhesion**					
Whey-derived MFGM (MFGM-10 Lacprodan^®^)	*L. rhamnosus* GG (LGG)	Male, 6-week-old BALB/c mice	Oral gavage of 0.1 mL MRSC media, MRSC with 5 g/L MFGM-10, MRSC with LGG (5 × 10^7^ CFU/mL) or MRSC with 5 g/L MFGM-10 and LGG (5 × 10^7^ CFU/mL) for 3 days	2.5 g/L MFGM-10 improved the survival of LGG in 0.5% bile in vitro potentially through modified EPS production Increased LGG viability after GI transit using cecum and fecal counts in mice	[87]
MFGM-derived MPL concentrate	*Lacticaseibacillus casei* OSU-PECh-C; *Lactobacillus acidophilus* Musallam2; *L. plantarum* subsp. *plantarum* TW14-1; *L. delbruekii* OSU-PECh-3	Gold (Au) Sensor; Caco-2/HT29-MTX	Examined the adhesion phenomena of 4 strains in the presence or absence of 0.5% (*w*/*v*) MPL to A) a gold sensor using a Quartz Crystal Micrograph with Dissipation (QCM-D); B) TEM; and C) intestinal cell culture	Binding properties observed by QCM-D and TEM suggest strain-specific differences in interactions with MPL *L. casei* and *L. delbruekii* exhibited greater adhesion to intestinal co-culture in the presence of 0.5% MPL	[88]
MFGM-derived MPL concentrate	*P. acidilactici* OSU-PECh-L; *P. acidilactici* OSU-PECh-3A; *L. plantarum* OSU-PECh-BB, *L. reuteri* OSUPECh-48; *L. casei* OSU-PECh-C, *L. paracasei* OSU-PECh-BA; *L. paracasei* OSU-PECh-3B	Caco-2	LAB strains grown with or without 0.5% (*w*/*v*) MPL were characterized by functional properties and their adhesive ability to fully differentiated Caco-2 cells	No change in autoaggregation or cell surface hydrophobicity in the presence of MPL 3 out of 7 strains showed increased adhesion to intestinal cells when grown in MPL	[86]
MFGM extract from butter serum	*L. rhamnosus* GG (LGG)	Caco-2 TC7	LGG was exposed to 5 mg/mL MFGM extract for 1 h and applied to intestinal cells (1 × 10^9^ CFU/mL)	Atomic force microscopy (AFM) demonstrated interactions between LGG and MFGM may be due to SpaCBA pili MFGM decreased LGG adhesion to intestinal epithelial cells	[76]
MPL-rich milk protein concentrate (Lacprodan^®^ PL-20)	*B. longum* subsp. *infantis* ATCC 15697	HT-29	Exposed bifidobacteria to MFGM ingredients for 1 h and measured adherence of bacteria to fully confluent cells after 2 h incubation using plate count method	PL-20 and BF decreased adhesion of ATCC 15697 MFGM-10 did not alter adhesion of ATCC 15697	[89]
Whey-derived MFGM (MFGM-10 Lacprodan^®^)					
Buttermilk fraction (BF)					
**Nutrient Absorption, Mucosal Immunity, and Gut Barrier Function**					
MFGM-derived MPL concentrate	*L. delbruekii* subsp. *bulgaricus* 2038; *S. thermophilus* 1131	Male Sprague-Dawley rats	Orally supplemented rats with SM, MPLs alone or either of these in fermented milk	Fermented milk containing SM increased serum ceramide levels twofold unlike standard milk MPLs in fermented milk also increased serum ceramide levels Unclear mechanism, but effect was attributed to a property of the fermented milk	[90]
Whey-derived MFGM (MFGM-10 Lacprodan^®^)	*L. paracasei* subsp. *paracasei* F19 (F19)	Infants (21-days−4-months old)	Double-blind RCT for an infant formula supplemented with MFGM (5 g/L prepared formula) or F19 (1 × 10^8^ CFU/L)	No differences in weight gain or growth between formula groups Compared to standard formula, both MFGM and F19 formulas resulted in fewer incidences of fever and days of fever more similar to breast milk (pre-registered primary outcome) No differences in respiratory tract infections between formula groups or breast-fed infants (pre-registered primary outcome)	[91]
Unspecified MFGM fraction	*B. animalis* subsp. *lactis* BB-12 (BB-12)	HT-29Cl34 (NF-κB reporter cell line); 28-day-old mice	Used reporter cell line to measure NF-κB activation in response to BB-12 (1 × 10^6^ or 1 × 10^7^ CFU/mL) and/or MFGM (50 μg/mL or 100 μg/mL) and LPS challenge (100 ng/mL); For in vivo mouse study, administered BB-12 (1 × 10^8^ CFU/day), MFGM (0.6 mg/g of body weight/day), or both BB-12 and MFGM orally for 1 or 4 weeks	BB-12 and MFGM treatment synergistically reduced LPS-provoked NF-κB activation in IECs greater than BB-12 or MFGM alone 4-week BB-12 and MFGM supplementation increased IgA-secreting mucosal B cell counts in Peyer’s patches, which further persisted 84 days after supplementation ended	[92]
**Neurodevelopment and Cognitive Function**					
MFGM components	*B. infantis* IM1; *L. rhamnosus* LCS-742	12-month-old infants	Double-blind RCT (COGNIS study) for a novel infant formula containing bioactive ingredients, including MFGM [10% of total protein content (*w*/*w*)] and probiotics	Improved brain maturation via cortical visual evoked potentials of infants fed experimental formula compared to standard formula at 12-months of age (pre-registered primary outcome) Responses of experimental formula-fed infants were closer to those of breast-fed infants	[93]
MFGM components	*B. infantis* IM1; *L. rhamnosus* LCS-742	4-year-old infants	Double-blind RCT (COGNIS study) for a novel infant formula containing bioactive ingredients, including MFGM [10% of total protein content (*w*/*w*)] and probiotics	Experimental formula improved language use and oral spontaneous expression at 4 years old by measure of the PLON-R test (pre-registered primary outcome)	[94]

## Data Availability

Not applicable.

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
