# Peer review of "Improving Human Health with Milk Fat Globule Membrane, Lactic Acid Bacteria, and Bifidobacteria"

_microorganisms, 2021, doi:10.3390/microorganisms9020341_

Round 1

Reviewer 1 Report

Both probiotics and MFGM have been shown as important for the proper functioning of the gut-brain axis. While there is published research on the combination of both, this topic has not been extensively reviewed and therefore this makes this manuscript original. However, it requires significant revisions to be suitable for publication.

The description of the MFGM is too general and does not take into account the compositional differences between species. There should be a special mention of human MFGM and its role for the infant, instead of a main focus on cow MFGM.

The sentence in lines 46-47 “During secretion, an additional layer of proteins forms between the outer bilayer and the monolayer” and the reference to the protein layer in Figure 1 are incorrect. There is no additional layer of proteins but the space between the monolayer and bilayer is electron-dense. MFGM proteins are co-located with phospholipids in the monolayer and bilayer.

Line 33: given concentrations for sphingomyelin are valid for human milk phospholipids but not for milk from cow and other dairy animals, which have lower levels.

Table 2: have all these gangliosides been identified in the MFGM? GD3 is the main one reported for cow’s milk.

Paragraph starting line 159: the role of LAB for infants is not much detailed and should be covered at least to the same extent as bifidobacteria. Some infant formulas contain LAB, the most common one is L. reuteri for the management of colic. Also it is necessary to discuss the human milk microbiome composition in this paragraph.

Figure 2: B1 and B2: what are the arrows pointing at?

Figure 3 and associated text in the manuscript are not easily understood and explained. Please add further information.

Lines 259-261: infant formulas are not autoclaved, so the sentence should be rephrased to make more sense. What was the MFGM used in this study? A specific dairy ingredient, an MFGM extract, an MFGM phospholipid extract…? This should always be made clear as many studies use phospholipid extract and therefore not the full set of MFGM components.

Many studies are covered. However the authors should clearly state when it is in vitro, ex vivo, in vivo animal and human clinical data that is reported.

Lines 320-322: please provide a reference.

Line 343: change to “64% of its adult size”

Lines 374-375: please change to “between formula-fed and breastfed infants”

I would recommend the authors to also consider the following points to strengthen the article:

  • There should be a section on bacterial survival in the food matrix containing MFGM
  • specific interaction between bacteria and MFGM in different food matrices (e.g. liquid vs powder)
  • list the specific strains that have been shown to interact with the MFGM
  • Identification or hypothesis on the key MFGM components that may interact with bacteria. Currently this is not well described and confusing at time sin the manuscript
  • Whether the MFGM structure and composition has an impact on interaction with bacteria: should the MFGM be located at the surface of the globule or separated MFGM in dairy products be as efficient?
  • Is the positive health effect due to biding of probiotic bacteria to MFGM for transport to site of action or the interaction of MFGM with other gut bacteria/pathogens promoting the growth of probiotic bacteria at play here?

References to add:

https://www.frontiersin.org/articles/10.3389/fped.2019.00347/full

https://www.nature.com/articles/s41598-019-47953-4

https://onlinelibrary.wiley.com/doi/abs/10.1002/mnfr.202000603

Author Response

Reviewer #1: Comments and Suggestions for Authors

Both probiotics and MFGM have been shown as important for the proper functioning of the gut-brain axis. While there is published research on the combination of both, this topic has not been extensively reviewed and therefore this makes this manuscript original. However, it requires significant revisions to be suitable for publication.

The description of the MFGM is too general and does not take into account the compositional differences between species. There should be a special mention of human MFGM and its role for the infant, instead of a main focus on cow MFGM.

R: We amended this issue by mentioning the composition of MPL’s of human milk in lines 133-138 and adding a new figure (Figure 2). In addition, we added lines 146-148 which mention the differences in MFGM proteins between human, bovine and caprine, as determined by proteomic analysis.

The sentence in lines 46-47 “During secretion, an additional layer of proteins forms between the outer bilayer and the monolayer” and the reference to the protein layer in Figure 1 are incorrect. There is no additional layer of proteins but the space between the monolayer and bilayer is electron-dense. MFGM proteins are co-located with phospholipids in the monolayer and bilayer.

R: We thought “interstitial protein layer” was better suited and more specific. To clarify this and make it more understandable, Figure 1 was amended as “interstitial layer” only and further description of this was added in lines 47-50. Certainly, the “space” between phospholipid monolayer and outer bilayer is cytoplasmic rich in electronic density, hereafter we named it interstitial layer. However, it is known to house at least XO/XDH and FABP and hence, we further named it interstitial protein layer. Indeed, XO/XDH is colocalized with integral membrane protein BTN from the moment they are synthetized in the apical part of the secretory cells (as shown in the insert of Figure 1, and experimentally observed in mice models in https://doi.org/10.1113/jphysiol.2002.027185) but not co-located with phospholipids. This argument would imply that both XO/XDH and FABP own a transmembrane domain in their structure. The three-dimensional structure of the XO/XDH (PDB: 1FOQ) has no such domain or even any phospholipid nor fatty acid binding site. In fact, this is deemed as a cytosolic protein found not only in the bovine MFGM fraction, but also in milk serum and whey (as reported experimentally in https://doi.org/10.1016/j.idairyj.2007.03.003); moreover, it is known to be associated with the small extracellular region of BTN protruding from the bilayer to the interstitial layer (as illustrated in Fig. 1B). Regarding FABP, which is also considered a cytoplasmic protein, and even though its three-dimensional structure has not been reported in the Protein Data Bank; a quick in silico analysis of the potential transmembrane domains based on sequences (UniProtKB: P48035 for bovine FABP4 and P55052 for bovine FABP5) showed no such domains, aside from the fatty acid binding sites. Due to these facts, an assumption that these proteins are co-located with membrane phospholipids would be wrong. References [6,7] was additionally amended.

Line 33: given concentrations for sphingomyelin are valid for human milk phospholipids but not for milk from cow and other dairy animals, which have lower levels.

R: We added a paragraph (lines 133-138).

Table 2: have all these gangliosides been identified in the MFGM? GD3 is the main one reported for cow’s milk.

R: Yes, we added lines 142-144 in this regard.

Paragraph starting line 159: the role of LAB for infants is not much detailed and should be covered at least to the same extent as bifidobacteria. Some infant formulas contain LAB, the most common one is L. reuteri for the management of colic. Also it is necessary to discuss the human milk microbiome composition in this paragraph.

R: We’ve added additional references to support the protective role of LAB in infants and information on the human milk microbiome composition (Lines 196-203).

Figure 2: B1 and B2: what are the arrows pointing at?

R: After changes, this figure is now called “Figure 3 B1 and B2”. Lines 463-464 state: “The thick white arrow shows the direction of the microscope stage travel”. This is typical in optical tweezer experiments.

Figure 3 and associated text in the manuscript are not easily understood and explained. Please add further information.

R: This information refers to the interaction of Lactobacillus and MFGM components in simple models; meaning not in dairy matrices directly. The information and images provided depict the association of two strains with MPL’s and the fact that some strains such as L. delbrueckii and L. plantarum accumulate MPL’s in the cell envelope, compared to their controls. Further information was added in lines 491-493.

Lines 259-261: infant formulas are not autoclaved, so the sentence should be rephrased to make more sense. What was the MFGM used in this study? A specific dairy ingredient, an MFGM extract, an MFGM phospholipid extract…? This should always be made clear as many studies use phospholipid extract and therefore not the full set of MFGM components.

R: We’ve improved the sentences for clarity (Lines 351-352), added the specific ingredients used to the text (Lines 346-347), and created a new table (Table 4), which summarizes this information and further details in the studies.

Many studies are covered. However, the authors should clearly state when it is in vitro, ex vivo, in vivo animal and human clinical data that is reported.

R: We modified the text appropriately and added an additional table (Table 4) with the model used for each study to help clarify this information.

Lines 320-322: please provide a reference.

R: We’ve added the references [92, 113] (line 438).

Line 343: change to “64% of its adult size”

R: This change has been implemented.

Lines 374-375: please change to “between formula-fed and breastfed infants”

R: This change has been implemented.

I would recommend the authors to also consider the following points to strengthen the article:

  • There should be a section on bacterial survival in the food matrix containing MFGM

R: Regarding bacterial survival, there is no current information due to the fact that food matrices containing LAB and MFGM are more focused on the compositional and sensory characteristics of the end product and the flavor development due to bacterial activity. This is a new subject that is gaining attention and research hasn’t been focusing on specifically the survival of bacteria in food matrices impacted by MFGM.

  • specific interaction between bacteria and MFGM in different food matrices (e.g. liquid vs powder)

R: Powder matrices are not consumed as such, but instead hydrated either previously or hydrated in the food matrix (i.e. skim milk powder or buttermilk powder added for yogurt or cheese manufacturing). Hence, the interaction would had taken place in the liquid form, and then dried if that’s the case (i.e. cheese powder). We don’t see the point in differentiating liquid vs. powder, but instead, mention that the water content in the matrix affects the interaction. In any case, and to clarify this point, we added a simple discussion in lines 234-239.

  • list the specific strains that have been shown to interact with the MFGM

R: We’ve added and made sure to list each strain as it is discussed in the text. In sections 1 and 2, we’ve listed these directly within the text. For Section 3, these strains are summarized in Table 4.

  • Identification or hypothesis on the key MFGM components that may interact with bacteria. Currently this is not well described and confusing at times in the manuscript

R: Studies reporting direct interaction of MFGM lipids with LAB were included in lines 481-325 (including Fig. 4). To further clarify this and mention the hypothetical interaction with MFGM glycoproteins, text was added in lines 473-481 and more information was added in lines 491-493.

  • Whether the MFGM structure and composition has an impact on interaction with bacteria: should the MFGM be located at the surface of the globule or separated MFGM in dairy products be as efficient?

R: There is no current information to address that issue and in addition, we’d need to define the binding “efficiency”, as you mention…how would you measure this? Physiochemically, biochemically, microscopically? As mentioned, there is no consensus up to this date. However, as shown in the literature reported here, LAB bind the MFGM located at the surface if the globule (Fig. 3A1, 3C1 and C2) and separated as a MPLs ingredient (Fig. 4A1 and B1). Lastly, the composition is likely to have an impact in the binding, but no information is available.

  • Is the positive health effect due to biding of probiotic bacteria to MFGM for transport to site of action or the interaction of MFGM with other gut bacteria/pathogens promoting the growth of probiotic bacteria at play here?

R: This is precisely why we think research in this area is important. Basically, we don’t know which is more important—the milk phospholipid interaction with the bacteria surface and its consequences on the intestine, or the general interaction of the microbiome when bacteria have been exposed and their metabolism change to change (or at least influence) the microbiome. We decided to address this in lines 513-518.

References to add:

https://www.frontiersin.org/articles/10.3389/fped.2019.00347/full

https://www.nature.com/articles/s41598-019-47953-4

https://onlinelibrary.wiley.com/doi/abs/10.1002/mnfr.202000603

R: Thank you for your suggestions. We’ve added the first reference to the manuscript in Table 4 and the remaining two references in lines 434-436.

Reviewer 2 Report

This review submission is generally well-written, polished and nicely structured. It deals with an interesting topic and summarises many aspects of the research well from the very small-scale biochemistry to the health implications. It also has the clear hypothesis that the way to advance MGFM is through the interaction with LAB and relatives

If I had to make one major criticism it would be that the health impacts can sound more conclusive than they are while some of the underlying studies are in model systems or has some design limitations. To address this I would argue for a table that summarises the purported health benefits in section 3, along with the study that supports this, model organism and experimental design (including if the outcomes mentioned are the pre-registered primary ones if an RCT)

Detailed points:

Table 3 could use more specific citations and some concentrations and amended with the trace proteins mentioned in the text

108: this reference is weird, how about that new lactobacillus nomenclature

172: read 45

174: this sentence is weird

187: strange argument

Section 3 should discuss the findings here https://pubmed.ncbi.nlm.nih.gov/30193112/ and the implications of how difficult it is to actually monitor colonization in vivo. 

289: don’t understand this reference

317: this whole paragraph should mention that this is done in piglets and the microbial compositional changes doesn’t translate. I think most infant maturation research would not argue that increasing protobacteria (e.coli) and decreasing firmicutes (faecalibacterium) would be helpful to infant gut maturation.

323: this section should specify what bacteria are in these patents

377: This kind of suggests that there has been factorial studies on the interactions of MGFM and microbiome, but I did not see any such studies. It’s obviously an interesting avenue though.

Perhaps 392-406 would be better suited earlier in the review while getting the broad overview.

Author Response

Reviewer #2: Comments and Suggestions for Authors

This review submission is generally well-written, polished and nicely structured. It deals with an interesting topic and summarises many aspects of the research well from the very small-scale biochemistry to the health implications. It also has the clear hypothesis that the way to advance MGFM is through the interaction with LAB and relatives

If I had to make one major criticism it would be that the health impacts can sound more conclusive than they are while some of the underlying studies are in model systems or has some design limitations. To address this, I would argue for a table that summarises the purported health benefits in section 3, along with the study that supports this, model organism and experimental design (including if the outcomes mentioned are the pre-registered primary ones if an RCT)

R: Thank you for your thoughtful suggestion. We created Table 4 to help summarize the ingredients, bacterial strains, models used, experimental design and key findings of each study.

Detailed points:

Table 3 could use more specific citations and some concentrations and amended with the trace proteins mentioned in the text

R: The protein profile of the MFGM is vast consisting of about 632 different proteins; moreover, according to proteomic studies we know that the distribution varies greatly upon the stage of lactation, environmental conditions influencing the cow, the method of extraction and the processing of the source containing the MFGM. We added a column with the known concentrations of MFGM proteins from raw cream and updated the text referring to Table 3 (lines 116-121) with more specific citations and the mention that the group of 8, are the predominant MFGM proteins. In addition, we also amended lines 102-105 to explain better.

108: this reference is weird, how about that new lactobacillus nomenclature

R: We have updated the reference (now listed as [48]) according the journal’s specifications. Additionally, we have implemented the new classification names of lactobacillus according to Zheng et al. 2020 (doi: 10.1099/ijsem.0.004107) throughout the paper.

172: read 45

 R: We changed the reference [47].

174: this sentence is weird

 R: We have revised the sentence.

187: strange argument

R: Amended by deleting it.

Section 3 should discuss the findings here https://pubmed.ncbi.nlm.nih.gov/30193112/ and the implications of how difficult it is to actually monitor colonization in vivo. 

R: Very cool paper. We’ve added it to the manuscript with a brief discussion in section 3.1 (Lines 407-414).

289: don’t understand this reference

R: We have updated the reference according to the instructions for authors for unpublished manuscripts (now listed as ref [88]) (https://www.mdpi.com/journal/microorganisms/instructions).

317: this whole paragraph should mention that this is done in piglets and the microbial compositional changes doesn’t translate. I think most infant maturation research would not argue that increasing protobacteria (e.coli) and decreasing firmicutes (faecalibacterium) would be helpful to infant gut maturation.

R: Absolutely. We’ve modified the paragraph to improve the clarity (lines 432-434) and added the reference [103].

323: this section should specify what bacteria are in these patents

R: We’ve added these strains both in the text (lines 441-444) and in the new Table 4.

377: This kind of suggests that there has been factorial studies on the interactions of MGFM and microbiome, but I did not see any such studies. It’s obviously an interesting avenue though.

 R: We have revised the sentence to eliminate suggestions of factorial studies (Line 500).

Perhaps 392-406 would be better suited earlier in the review while getting the broad overview.

R: Thank you for the suggestion. We have moved the paragraph to Section 2 Lines 220-232.

Round 2

Reviewer 1 Report

Thank you for the answers and appropriate changes to the manuscript.